# OpenReview forum: "Teaching LLMs to Decode Activations Into Natural Language"
_ICLR.cc/2025/Conference — Submitted to ICLR 2025_

### Official Review · Reviewer_eoci · 2024-10-29

**Soundness:** 3
**Presentation:** 2
**Contribution:** 3
**Rating:** 5
**Confidence:** 3

**Summary:**

This paper introduces **LatentQA**, a novel interpretability task designed to translate large language model (LLM) activations into natural language, making model internals more accessible and understandable for humans. Unlike traditional interpretability techniques that yield abstract outputs (e.g., vectors or circuits), LatentQA enables direct, human-readable interpretations of model behavior. To achieve this, the authors constructed a comprehensive LatentQA dataset, created with the aid of GPT-4 to generate question-answer pairs that describe qualitative attributes of model activations. They further developed a method called **Latent Interpretation Tuning (LIT)** to fine-tune an LLM to perform LatentQA, effectively enabling it to "caption" model activations with descriptive language. This approach not only advances interpretability but also offers a means to control model behaviors, such as reducing biases and steering sentiment, through its differentiable, language-based output.

**Strengths:**

1. The proposed LatentQA task introduces a groundbreaking method for interpreting LLM activations by translating them directly into natural language. This approach is more user-friendly and accessible compared to existing interpretability techniques, which often rely on complex, abstract outputs.

2. LIT uniquely combines interpretability with control, enabling it not only to decode and explain LLM activations but also to steer model outputs in desired directions, such as adjusting biases or modifying response tones.

3. LIT offers a powerful approach for identifying and revealing potentially harmful model behaviors, making it a valuable tool for safety auditing and alignment.

**Weaknesses:**

1. The paper would benefit from an improved pipeline diagram to clarify the process flow.

2. LatentQA heavily relies on extensive data and faces challenges in scaling efficiently. The model’s interpretative ability is limited to the specific personas and activations it was trained on, which restricts its generalizability. For example, it can only interpret personas included in its training dataset. It would be valuable to assess whether LatentQA can perform well in a zero-shot setting, such as on tasks similar to those in [1].

3. The paper states that LatentQA selects the 15th layer's activations as input for the decoder, supported by early experimental results. However, intuitively, activations from later layers in large language models may be better learned and thus more effective. An ablation study analyzing the impact of different layers’ activations as input would enhance the robustness of the findings.

4. The dataset used is synthetically generated, which could introduce biases or noise. A more detailed description of the dataset construction process, along with an analysis of its diversity, would strengthen the paper's validity.

[1] Hendel, Roee, Mor Geva, and Amir Globerson. "In-context learning creates task vectors." arXiv preprint arXiv:2310.15916 (2023).

**Questions:**

Will the data used in this study be publicly accessible?

---

> ### Author Response · Authors · 2024-12-02
>
> Thank you for your thorough review and helpful suggestions. We have implemented them and believe that they will help improve the quality of the paper.
>
> *The paper would benefit from an improved pipeline diagram to clarify the process flow.*
> - We have updated Figure 1 and included a new Figure 2 to clarify the process flow.
> - Please let us know if you have additional suggestions to the figures, as we would be excited to make them more legible.
>
> *LatentQA heavily relies on extensive data and faces challenges in scaling efficiently.*
> - We refer the reviewer to the common response part 1 (scaling experiments). We paraphrase the main takeaway of Section 5.3: Latent Interpretation Tuning scales with both increasing dataset size and increasing model size, which demonstrates the power of training-based approaches.
>
> *The model’s interpretative ability is limited to the specific personas and activations it was trained on, which restricts its generalizability. For example, it can only interpret personas included in its training dataset. It would be valuable to assess whether LatentQA can perform well in a zero-shot setting*
> - We note that all of the personas and questions we interpret in the reading and control sections (5.1 and 5.2) are novel, which suggests that our decoder is indeed able to generalize.
> - Finally, we refer the reviewer to the common response part 1 (additional control experiments). For convenience, we paraphrase:
>   - We are able to control the model to respond differently to user questions depending on the model’s belief of the user’s demographic information, e.g., if the user asks “What car should I purchase?”, the model’s response will differ depending on their income.
>   - This suggests that the decoder is able to generalize to novel types of control at test-time.
>
> *An ablation study analyzing the impact of different layers’ activations as input would enhance the robustness of the findings.*
> - We refer the reviewer to the common response part 2 (reading and writing layer ablation). We find that our initial choice of layers (layer $k = 15$ to read from and layer $\ell=0$ to write to) yields the best generalization across our hyperparameter study.
> - Intuitively, this result makes sense for two reasons. First, we read from layer $k=15$ because we expect the middle layers of the model to be the most semantically rich [1]. Second, we write to layer $\ell = 0$ because we want to provide the decoder with the most steps to process the activation. Although there is a distribution shift from the 15th to the 0th layer, the decoder is trained, so it learns to handle this distribution shift.
>
> *A more detailed description of the dataset construction process, along with an analysis of its diversity, would strengthen the paper's validity.*
> - We have rewritten Section 3 to more precisely discuss the dataset creation. We have also added Appendix A, which provides the prompts and more thoroughly discusses the process used to create our dataset construction process. We are happy to include more specific information if you have any suggestions.
> - Could you clarify what you mean by an analysis of the dataset’s diversity? We are happy to run more detailed analyses.
>
> *Will the data used in this study be publicly accessible?*
> - Yes. We will release the data after the anonymity period.

---

> ### Author Response · Authors · 2024-12-04
>
> Dear Reviewer eoci,
>
> Thank you for your detailed and insightful suggestions! We hope that our previous comment and the revised version of the paper (available on OpenReview) have adequately addressed your concerns.
>
> If the revisions meet your expectations, we kindly ask if you might consider updating your review to reflect your revised opinion of the paper.

---

### Official Review · Reviewer_XCEP · 2024-10-29

**Soundness:** 2
**Presentation:** 2
**Contribution:** 3
**Rating:** 5
**Confidence:** 3

**Summary:**

The authors propose the task and dataset for LatentQA -- a set of ([ACT], question, answer), where [ACT] is the model activation when prompted with the question and answer. This question and answer is attributed with stimuli -- something like a persona, or style to answer the question. For example, given an original question "What should I have for dinner?" the authors add a stimuli like "Answer as if you were a billionaire living on a private island", and get the model's answer by: original_question + stimuli. Next, the authors ask GPT-4 to generate QA pairs to describe the properties of the model's answer.

This LatentQA dataset is then used to train a decoder to produce the answer given the question, and patching [ACT] to the decoder.

Next, this decoder can be used to read/control LLM behavior. The experimental result is based on this read and control task

**Strengths:**

- An important and novel problem: how to interpret LLM activation in natural language
- Clever way to materialize this motivation into an idea
- I like that the resulting decoder can be used to both interpret activation and control LLM behavior

**Weaknesses:**

Execution of a great idea that falls short. For several reasons:
1. I find patching activation from layer 15 to layer 0 weird. Wouldn't the model just completely fail? When you get layer 15's activation, it is already accumulating outputs from layers 0-14, and thus numerically will be very different from the original distribution of the model's layer 0 activation. I also fail to understand the reasoning behind doing this.
2. The method is model specific -- so each time a person wants to do this for a different model, they need to collect 1.2 data points from that model? they can probably still use the same QA provided by the original LatentQA dataset, but it is still a heavy price.
3. Some crucial writing needs to be made more clear (check Questions)
4. On 5.1 results: I do not find this evaluation fair (comparing training-free method with training-based method); are there other training-based methods you can compare to?
5. On 5.1: There is a debiasing specific activation steering method: https://arxiv.org/pdf/2406.03631 have you tried to compare debiasing results with this method?
6. On 5.2: LIT seems to perform the worst on 'Generate Positive'.

**Questions:**

1. Is dialog = control + control response? (lines 214-215)
2. What do you mean by repeatedly calling STEER(ACT, "control")? (lines 258-259)
3. Is STEER(ACT, "control") the gradient computed when doing LoRA fine-tuning on the "control" question-answer string, and patching the activation with [ACT]?
4. Why do you need to sample activations from prompts from he Databricks Dolly dataset on top of the stuffs you already have on LatentQA?
5. In my understanding, RepE is a fully training free, representation engineering method. Why did you use LoRRA fine-tuning with RepE?

---

> ### Author Response · Authors · 2024-12-02
>
> Thank you for your helpful suggestions. We have implemented them.
>
> *I find patching activation from layer 15 to layer 0 weird.*
>
> - We refer the reviewer to the common response part 2 (reading and writing layer ablation). We find that our initial choice of layers (layer $k = 15$ to read from and layer $\ell=0$ to write to) yields the best generalization across our hyperparameter study.
> - Intuitively, this result makes sense. First, we read from layer $k=15$ because we expect the middle layers to be the most semantically rich. Second, we write to layer $\ell = 0$ because we want the decoder to have the most steps to process the activation. Although there is a distribution shift from the 15th to the 0th layer, the decoder is trained, so it learns to generalize.
>
> *so each time a person wants to do this for a different model, they need to collect 1.2 data points from that model?*
>
> Yes, although this is not a substantial cost.
> - We have improved our dataset (Section 3) and find in practice we are able to achieve similar performance with only 17K data points. Our initial dataset during submission had many duplicated data points, and we find that removing these in fact improves performance.
> - At its current size, our dataset is smaller than most instruction-tuning datasets (e.g., Alpaca has 52K instructions) and trains faster.
> - Finally, we note that training a new system for each type of LLM backbone is a common paradigm. For example, there is a [new copy of LLaVA trained for each LLM](https://github.com/haotian-liu/LLaVA/blob/main/docs/MODEL_ZOO.md).
>
> *On 5.1 results: are there other training-based methods you can compare to?*
>
> Yes; we refer the reviewer to common response part 3 (benchmarking against trained baselines). For convenience, we paraphrase:
> - For the first reading task, we add linear probing, a baseline trained on task-specific data and find that our decoder outperforms linear probing by an absolute 32.3% over 6 tasks.
> - For the second reading task, linear probing is not applicable because we do not have a prior knowledge of the system prompts to probe. Unfortunately, there are no other appropriate trained baselines.
> - Finally, we compare our decoder to all pre-existing LatentQA systems, neither of which train their decoder. We view part of our contribution as the first method to train a system for LatentQA.
>
> *On 5.1: There is a debiasing specific activation steering method... have you tried to compare debiasing results with this method?*
>
> The method in the listed paper is customized for position bias (whether the model assigns higher likelihood to the first or last answers in a prompt), whereas the CrowS pairs dataset focuses on stereotype bias (whether the model assigns higher likelihood to sentences with stereotypes). We are not aware of a simple way to convert that method to our application, but would be happy to implement suggestions.
>
> *On 5.2: LIT seems to perform the worst on 'Generate Positive'.*
>
> LIT performs worse than RepE, however, it still outperforms prompting. We will more carefully remark upon this limitation in the final revision, as it suggests future work may help improve the robustness of the control.
>
> ## *Writing clarifications*
> *Is dialog = control + control response?*
> - Yes; moreover, we have updated sections 3 and 4 to make their notation more consistent.
>
> *What do you mean by repeatedly calling STEER(ACT, "control") and what is STEER(ACT, “control”)?*
> - We rewrote the explanation of control in Section 4 and in Appendix B.3. Roughly,  STEER([Act], control) is the gradient with respect to [Act] of the decoder’s logprob of generating answer given [Act] + question. By repeatedly updating [Act] with these gradients, we can approximately identify the target LLM's activations that maximizes the probability of the control.
>
> *Why do you need to sample activations from prompts from he Databricks Dolly?*
> - When performing the control, we need activations from the target LLM. During training, these activations come from stimulus prompts. During control, we need some set of stimulus prompts. Most sets of stimulus prompts (as long as the prompts are diverse) work for control. We thus need a diverse bank of prompts, e.g., Databricks Dolly dataset.
>
> *In my understanding, RepE is a fully training free, representation engineering method. Why did you use LoRRA fine-tuning with RepE?*
> - The representation engineering (RepE) paper describes two methods for control: a training-free method, which adds steering vectors to guide activations, and a training-based method, which trains a LoRA to approximate the effect of adding steering vectors to activations. The latter training-based method is called LoRRA finetuning.
> - Because we want a fair comparison to a trained method, we compare our decoder to RepE using LoRRA finetuning. See [1] for more details.
>
> [1] Zou, Andy, et al. "Representation engineering: A top-down approach to ai transparency." arXiv preprint arXiv:2310.01405 (2023).

---

> ### Comment · Reviewer_XCEP · 2024-12-03
>
> Thank you for the rebuttal and clarification. My concerns are mainly addressed. To summarize, i still think the paper has great potential, and the revision made by the authors greatly improve the paper. I still think the results for control experiment is a little shy -- I will raise my score to 5 for now. For future iterations, it would be great if we can see more and better results for control experiments. Perhaps its more suitable for other control tasks -- jailbreak a safety aligned model and safeguard a base model?
>
> Another thing that will strengthen the paper is more extensive evaluation on the jailbreaking control... there is a comprehensive benchmark on jailbreaking like MaliciousInstruct dataset, JailBreakBench dataset.. or you can use a toxicity-based dataset. The evaluation can be performed using win rate \%or toxicity score.. rather than the current manual inspection

---

### Official Review · Reviewer_RaEp · 2024-11-03

**Soundness:** 3
**Presentation:** 3
**Contribution:** 3
**Rating:** 6
**Confidence:** 3

**Summary:**

In this work, the authors explore an interesting problem of LLMs to interpret their own latent representations (activations). They curated a dataset called LATENTQA with over 1M QA pairs and used it to train a decoder specifically designed for this purpose. The authors demonstrate a range of valuable applications for their proposed model, including reading model latent representations and controlling model behavior. Compared to existing methods, the trained decoder achieves a significant performance improvement.

**Strengths:**

1. The studied problem, directly decoding activations into natural language,  is interesting and unique.

2. They contribute a unique large-scale dataset with 1.2 million data points specifically designed for this task, which is a valuable resource for the community.

3. The authors investigate a diverse range of applications for their approach, demonstrating its effectiveness across several tasks. These include extracting relational information from latent representations, decoding personas embedded in system prompts, controllable sentiment generation, debiasing, and eliciting harmful capabilities.

4. The proposed method shows promising results across these applications.

**Weaknesses:**

1. For the experiments, there’s a concern about fairness in comparison. Since the proposed model is trained on a large-scale dataset, many of the baselines used are training-free methods. Comparing these directly may not be entirely fair, and it may lead to overclaiming the observed performance improvements.

2. Another concern is the generalization capability of the model. Specifically, it would be insightful to test whether it’s possible to achieve cross-model understanding, such as using activations from one model (e.g., LLaMA) to interpret the behavior of a different model (e.g., Mistral).

3. The authors frequently mention "interpretability," but the connection between this approach and interpretability is not entirely clear. It’s challenging to see how the studied applications, such as decoding latent information or controlling behaviors, directly contribute to a deeper understanding of the model’s internal workings.

**Questions:**

Here is a list of questions for the authors:

1. **Practical Use of Predicting Future Model Behavior**: What are the practical applications of predicting future model behavior based on latent representations, and why is this an important problem to investigate?

2. **Contextualization Using WikiText-103**:
   - (a) Since WikiText-103 is used to extract phrases containing the subjects, these phrases may also include objects. Could this lead to potential data leakage?
   - (b) Do other baselines also use WikiText-103 for contextualization? If not, could this create an unfair comparison?

3. **Determining the Appropriate Layer for Activations**: In the footnote, it mentions "during execution replace the activations of ??? with [ACT] at the appropriate layer." How is the “appropriate layer” determined for replacing activations? Section 4 notes that the activations of the 15th layer in the target model are used as input, based on early experiments. Could you provide more detail on how these early experiments guided the choice of layer?

4. **Choice of Layer for Replacement**: Why do you consistently replace activations at layer 0, rather than other layers? Is this choice shown to be optimal, or were other layers considered as well? Has any analysis been conducted to evaluate whether this layer is indeed the most effective for achieving the desired outcomes?

These clarifications would help in understanding the setup and methodology more thoroughly.

**Details Of Ethics Concerns:**

On line 462, the authors suggest that the proposed method has the potential to generate harmful content across various domains, including the creation of recipes for bioweapons and cyberweapons. Although the authors argue that their use of publicly available models (e.g., Llama 3) is unlikely to significantly impact the public’s capacity to develop harmful models, an additional ethics review could still be beneficial.

---

> ### Author Response · Authors · 2024-12-02
>
> Thank you for your thorough review and helpful suggestions. We have implemented them and believe that they will help improve the quality of the paper.
>
> *Comparing these directly may not be entirely fair, and it may lead to overclaiming the observed performance improvements.*
>
> We refer the reviewer to the common response part 3 (benchmarking against trained baselines). For convenience, we paraphrase:
> - For the first reading task, we add linear probing, a baseline trained on task-specific data and find that our decoder outperforms linear probing by an absolute 32.3% over 6 tasks.
> - For the second reading task, linear probing is not applicable because we do not have a prior knowledge of the system prompts to probe.
> - In all of the control tasks, we already compare to a trained baseline, the LoRRA finetuning method from RepE.
> - Finally, we compare our decoder to all pre-existing LatentQA systems, neither of which train their decoder. Our method outperforms both.
>
> *Another concern is the generalization capability of the model. Specifically, it would be insightful to test whether it’s possible to achieve cross-model understanding, such as using activations from one model (e.g., LLaMA) to interpret the behavior of a different model (e.g., Mistral).*
>
> We refer the reviewer to the common response part 1 (additional control experiments). For convenience, we paraphrase:
> - We are able to control the model to respond differently to user questions depending on the model’s belief of the user’s demographic information, e.g., if the user asks “What car should I purchase?”, the model’s response will differ depending on their income. Our decoder performs as well as prompting and linear probing.
> - This suggests that the decoder is able to generalize to novel types of control at test-time.
> - We will post results demonstrating that our LatentQA training applies to other models, such as Mistral’s Ministral-8B-Instruct-2410.
> - We agree that your experiment would be interesting to test. However, a more natural setup would be to train a new decoder for each target LLM, similar to visual instruction tuning (e.g., there are multiple versions of LLaVA, one for each LLM).
>
> *It’s challenging to see how the studied applications, such as decoding latent information or controlling behaviors, directly contribute to a deeper understanding of the model’s internal workings.*
>
> - Thank you for pointing out this inconsistency. Our work focuses more on applications of interpretability, specifically, developing techniques that enable novel applications, such as predicting future model behavior from the activations.-
> - We rewrote the introduction to more carefully describe our focus on applications of interpretability, rather than understanding the minutiae of LLM internals (which is more akin to mechanistic interpretability).
>
> *What are the practical applications of predicting future model behavior based on latent representations, and why is this an important problem to investigate?*
> - An important practical application we hope LatentQA addresses is monitoring of harmful model behavior. In particular, models exhibit unexpected harmful behaviors at test time if their goals change, e.g., through a feedback loop [1] or with a prompt injection [2]. A LatentQA system could help interpret the internal state of the target LLM and flag any potentially erratic or harmful behavior, serving as an additional guardrail on model safety.
>
> *Since WikiText-103 is used to extract phrases containing the subjects, these phrases may also include objects. Could this lead to potential data leakage?*
> - Thank you for the insightful observation. The dataset was filtered to remove the objects. We will note this in our dataset construction.
>
>
> *How is the “appropriate layer” determined for replacing activations? Why do you consistently replace activations at layer 0, rather than other layers?*
> - We refer the reviewer to the common response part 2 (reading and writing layer ablation). We find that our initial choice of layers (layer $k = 15$ to read from and layer $\ell=0$ to write to) yields the best generalization across our hyperparameter study.
> - Intuitively, this result makes sense for two reasons. First, we read from layer $k=15$ because we expect the middle layers of the model to be the most semantically rich [3]. Second, we write to layer $\ell = 0$ because we want to provide the decoder with the most steps to process the activation. Although there is a distribution shift from the 15th to the 0th layer, the decoder is trained, so it learns to handle this distribution shift.
>
> [1] Pan, Alexander, et al. "Feedback loops with language models drive in-context reward hacking." arXiv preprint arXiv:2402.06627 (2024).
>
> [2] Wu, Chen Henry, et al. "Adversarial Attacks on Multimodal Agents." arXiv preprint arXiv:2406.12814 (2024).
>
> [3] Ghandeharioun, Asma, et al. "Who's asking? User personas and the mechanics of latent misalignment." arXiv preprint arXiv:2406.12094 (2024).

---

> ### Author Response · Authors · 2024-12-04
>
> Dear Reviewer RaEp,
>
> Thank you for your detailed and insightful suggestions! We hope that our previous comment and the revised version of the paper (available on OpenReview) have adequately addressed your concerns.
>
> If the revisions meet your expectations, we kindly ask if you might consider updating your review to reflect your revised opinion of the paper.

---

### Official Review · Reviewer_zR7e · 2024-11-03

**Soundness:** 2
**Presentation:** 3
**Contribution:** 2
**Rating:** 5
**Confidence:** 3

**Summary:**

The paper introduces a new task LatentQA. It aims at improving interpretability and control in language models (LLMs) by transforming latent activations into language responses. To facilitate this, the authors propose Latent Interpretation Tuning (LIT), which fine-tunes a decoder model to answer open-ended questions about model activations. By doing so, the decoder can provide insights into latent model tendencies, biases, and behavioral control. Authors provide comprehensive and various tasks for the experimental setting and showcase the advanced model performance.

**Strengths:**

1. This paper propose a new dataset, LatentQA, aiming to improve interpretability and control in language models (LLMs) by transforming activations into language responses.
2. This paper includes comprehensive experiments and anlysis
3. This paper provides some insights from explaining activations in LLM.

**Weaknesses:**

1. This paper is hard to follow at the first glance. The term 'activation' should be explained clearer at the beginning.
2. When comparing with Patchscope, it would be fair and convincing to use the same LLaMA3.1-8b in your method for Patchscope.
3. In the controllable generation, control refers to persona, I would like to see other types of control in the experiment, e.g., topics
4. Methods like DExpert and Patchscope are training-free methods, while your LIT needs finetuning LLMs. The comparison seems to be unfair. At least you should compare the training/inference latency between different strategies.
5. There's only one LLM employed for the task, I would expect to see more LLMs to be tested.

I would be happy to raise my score once my concerns are addressed.

**Questions:**

1. What is LoRRA finetuning?

---

> ### Author Response · Authors · 2024-12-02
>
> Thank you for your review and detailed suggestions. We have implemented them and believe that they will help improve the quality of the paper.
>
> *“The term ‘activation’ should be explained clearer at the beginning*
> - We have changed the description of LatentQA to be more precise. Particularly, from L38-L40: “we consider the task of LatentQA, open-ended question answering (QA) about latents, i.e., model activations, in natural language.”
> - In particular, we define latents as model activations. Please let us know if you have further clarity suggestions.
>
> *It would be fair and convincing to use the same Llama-3 when comparing to Patchscope*
> - We reran the Patchscope experiment with Llama-3-8B-Instruct and updated Table 1 with the new numbers. They are reproduced below for your convenience. In particular, we see that our decoder still outperforms Patchscope and linear probing, a trained baseline that requires task-specific data.
>
> | Method | Country_Curr | Food_Country | Ath_Position | Ath_Sport | Prod_Company | Star_Const |
> |---------|--------------|--------------|--------------|------------|--------------|------------|
> | Linear Probe | 17.7 ± 2.2 | 5.1 ± 3.7 | 75.9 ± 9.1 | 53.8 ± 10.3 | 58.9 ± 7.2 | 17.5 ± 5.3 |
> | Patchscope | 24.3 ± 2.3 | 36.2 ± 3.8 | 51 ± 2 | 28.9 ± 1.4 | 28 ± 1.8 | 24.6 ± 1.6 |
> | Ours | **86.9 ± 1.0** | **68.9 ± 2.0** | 65.2 ± 2.2 | **90.4 ± 0.8** | **71.5 ± 4.8** | **39.2 ± 4.2** |
>
> *I would like to see other types of control in the experiment, e.g., topics*
>
> - We refer the reviewer to the common response part 1 (additional control experiments). For convenience, we paraphrase:
> - We are able to control the model to respond differently to user questions depending on the model’s belief of the user’s demographic information, e.g., if the user asks “What car should I purchase?”, the model’s response will differ depending on their income. Our decoder performs as well as prompting and linear probing.
> - This suggests that the decoder is able to generalize to novel types of control at test-time.
>
> *Methods like DExpert and Patchscope are training-free methods, while your LIT needs finetuning LLMs. The comparison seems to be unfair.*
> We refer the reviewer to the common response part 3 (benchmarking against trained baselines). For convenience, we paraphrase:
> - For the first reading task, we add linear probing, a baseline trained on task-specific data and find that our decoder outperforms linear probing by an absolute 32.3% over 6 tasks.
> - For the second reading task, linear probing is not applicable because we do not have a prior knowledge of the system prompts to probe.
> - In all of the control tasks, we already compare to a trained baseline, the LoRRA finetuning method from RepE.
> - Finally, we compare our decoder to all pre-existing LatentQA systems, neither of which train their decoder. Our method outperforms both.
>
> *I would expect to see more LLMs to be tested.*
> - We are currently running the reading experiments (Section 5.1) with Mistral’s Ministral-8B-Instruct-2410 and will post the results after they finish.
> - We are also happy to run experiments with additional language models for the final revision.
>
> *What is LoRRA finetuning?*
> - We updated the paper to have a more precise definition: We compare to RepE (Zou et al., 2023), which has two methods of control: a training-free method, which adds steering vectors to activations, and a training-based method, which updates weights to approximate adding steering vectors. For RepE, we use the training-based method (called LoRRA finetuning) for a fair comparison.
> - Please let us know if you have further questions.

---

> > ### Comment · Reviewer_zR7e · 2024-12-03
> >
> > Great, thanks for your further clarification. My concerns are partially resolved. However, I still expect to see more model perforance in the paper. I'll raise my score to 5.

---

### Author Response · Authors · 2024-12-02
**Common response part 1: updated paper and additional experiments**

## Updated paper
To improve the quality of our paper, we have made the following changes (which can be viewed in the revised version):
- Updated clarity of section 1 (intro), section 3 (dataset), and section 4 (method). Added Appendix A and B outlining sections 3 and 4, respectively, in more detail.
- Combined the previous section 5.3 (harmful capability elicitation) into section 5.2.
- Added a new section 5.3 that discusses scaling LatentQA systems.
## Scaling LatentQA systems
In section 5.3, we demonstrate that our training method scales with both dataset size and model size. This suggests that training LatentQA systems will improve with scale. We view part of our contribution as demonstrating the importance of training the decoder, as all prior LatentQA systems ([1] and [2]) use an untrained decoder.
## Additional control experiments demonstrating generalization
Reviewers zR7e, RaEp, and eoci were interested in the generalization capabilities of the decoder. The reading tasks (section 5.1) demonstrate the decoder’s ability to read novel activations and questions, as none of the questions nor tasks were in our training dataset. To further demonstrate the decoder’s ability to perform novel types of control, we run a new experiment, following the setup from [3].

In particular, given a specific user demographic attribute (e.g., high socioeconomic status), our goal is to control the target LLM’s response to open-ended questions (e.g., “What car should I buy?”) based on the user’s demographic attribute (e.g., “You should consider a luxury sedan”). We measure the number of responses that are successfully steered according to the user’s demographic attribute using GPT-4o. Our questions, all of which come from [3], fall into four categories: education level (uneducated vs. educated), socioeconomic status (high vs. low), age (young vs. old), and gender (male vs. female).

We report results in the below table, comparing the percent of successfully steered questions (out of 30 total) between using our decoder, linear probing (a trained baseline), and prompting. We see that our decoder is able to generalize and steer model’s beliefs of user demographic attributes, even though this type of control was not seen during training. We are happy to expand this experiment for the final revision if the reviewers believe it to be important to the paper’s message.

| Method | Age | Gender | Education | Socioeconomic status|
|---------|--------------|--------------|--------------|------------|
| Linear Probe | 100 | 93 | 100 | 97 |
| Prompting | 100 | 100 | 100 | 100 |
| Ours | 100 | 100 | 100 | 100 | 100 |


[1] Ghandeharioun, Asma, et al. "Patchscope: A unifying framework for inspecting hidden representations of language models." arXiv preprint arXiv:2401.06102 (2024).

[2] Chen, Haozhe, Carl Vondrick, and Chengzhi Mao. "Selfie: Self-interpretation of large language model embeddings." arXiv preprint arXiv:2403.10949 (2024).

[3] Chen, Yida, et al. "Designing a Dashboard for Transparency and Control of Conversational AI." arXiv preprint arXiv:2406.07882 (2024).

---

> ### Author Response · Authors · 2024-12-02
> **Common response part 2: ablation of reading and writing layers**
>
> Reviewers RaEp, XCEP, and eoci were interested in how we determined which layer $k$ to read activations from and which layer $\ell$ to write activations to. To robustly determine the best layers to use, we performed a hyperparameter sweep over $k, \ell \in { 0, 7, 15, 22, 30}$ with Llama-3-8B-Instruct. We train on the same dataset obtained in Section 3.
>
> Performance of each checkpoint was calculated using the evaluation set detailed in Section 5.1. We find that our layers $k = 15$ and $\ell = 0$ attains the lowest eval loss, suggesting our original choice of layers has the best LatentQA generalization.
>
> Intuitively, this result is sensible: we read from the middle layers because they contain the most semantically-rich representations [1] and we write to the 0th layer because we want to provide our decoder with as many steps for processing the activation as possible. Furthermore, the decoder is trained, so it learns to handle the distribution shift from layer $k = 15$ to layer $\ell = 0$.
>
>
> | Read Layer ($k$) | Write Layer ($\ell$) = 0 | Write Layer ($\ell$) = 7 | Write Layer ($\ell$) = 15 | Write Layer ($\ell$) = 22 | Write Layer ($\ell$) = 30 |
> |----------------|---------------------|---------------------|----------------------|----------------------|----------------------|
> | 0              | 1.165               | 1.277               | 1.374                | 1.435                | 1.564                |
> | 7              | 1.114               | 1.024               | 1.119                | 1.222                | 1.311                |
> | 15             | **1.013**           | 1.017               | 1.076                | 1.171                | 1.269                |
> | 22             | 1.039               | 1.049               | 1.102                | 1.168                | 1.268                |
> | 30             | 1.067               | 1.084               | 1.129                | 1.176                | 1.261                |
>
> [1] Ghandeharioun, Asma, et al. "Who's asking? User personas and the mechanics of latent misalignment." arXiv preprint arXiv:2406.12094 (2024).

---

> > ### Author Response · Authors · 2024-12-02
> > **Common response part 3: benchmarking against trained baselines**
> >
> > Reviewers zR7e, RaEp, and XCEP were all interested in adding additional trained baselines to our experiments. We first note that we compared to all pre-existing LatentQA baselines. In particular, there are (to our knowledge) only two prior works which build LatentQA systems ([1] and [2]), but both use an untrained decoder. Therefore, we view part of our contribution as demonstrating the importance of training the decoder for building LatentQA systems. An added benefit is that unlike training-free methods for LatentQA systems, our latent interpretation tuning improves with more increasing dataset size and increasing model size (Section 5.3).
> >
> > Moreover, we have updated the paper and now we compare our decoder to trained baselines whenever possible. In particular:
> > - **Reading experiments**:
> >   - For the task of extracting information from latent representations, we redo Table 1 to include linear probing, a trained baseline on task-specific data. We see, however, that our decoder still outperforms the trained linear probe baseline by an absolute difference of 32.2% over 6 tasks. We reproduce Table 1 below for convenience.
> >   - For the task of uncovering hidden system prompts, linearing probing is the only possible trained baseline for this task. Linear probes require a priori knowledge of the concept being probed. By construction, however, we do not have access to that in this task, so we do not compare our decoder to linear probing.
> > - **Control experiments**:
> >   - For all three control experiments, we do compare to a trained baseline. In particular, in each of the tasks we compare to the RepE baseline, which is trained. The representation engineering paper [3] has two methods for controlling models: a training-free method, which adds steering vectors to guide activations, and a training-based method, which trains a LoRA to approximate the effect of adding steering vectors to activations. For a fair comparison, in our paper, we do compare our decoder to RepE’s training-based method, which is called LoRRA finetuning. See [3] for more details.
> >
> > | Method | Country_Curr | Food_Country | Ath_Position | Ath_Sport | Prod_Company | Star_Const |
> > |---------|--------------|--------------|--------------|------------|--------------|------------|
> > | Linear Probe | 17.7 ± 2.2 | 5.1 ± 3.7 | 75.9 ± 9.1 | 53.8 ± 10.3 | 58.9 ± 7.2 | 17.5 ± 5.3 |
> > | Patchscope | 24.3 ± 2.3 | 36.2 ± 3.8 | 51 ± 2 | 28.9 ± 1.4 | 28 ± 1.8 | 24.6 ± 1.6 |
> > | Ours | **86.9 ± 1.0** | **68.9 ± 2.0** | 65.2 ± 2.2 | **90.4 ± 0.8** | **71.5 ± 4.8** | **39.2 ± 4.2** |
> >
> >
> > [1] Ghandeharioun, Asma, et al. "Patchscope: A unifying framework for inspecting hidden representations of language models." arXiv preprint arXiv:2401.06102 (2024).
> >
> > [2] Chen, Haozhe, Carl Vondrick, and Chengzhi Mao. "Selfie: Self-interpretation of large language model embeddings." arXiv preprint arXiv:2403.10949 (2024).
> >
> > [3] Zou, Andy, et al. "Representation engineering: A top-down approach to ai transparency." arXiv preprint arXiv:2310.01405 (2023).

---

### Meta-Review · Area_Chair_BT53 · 2024-12-14

**Metareview:**

This paper introduces LatentQA, which aims at improving interpretability and control in LLMs by transforming latent activations into language responses. After author rebuttal, it received scores of 5556. On the one hand, all the reviewers agree that the studied problem, directly decoding activations into natural language, is interesting and unique. On the other hands, several major concerns have been raised, including (1) paper writing clarify can be improved, (2) generalization of the proposed method, (3) ablation of how the reading and writing layers are selected, and (4) benchmarking against baselines with model training, instead of comparing with training-free methods, which seems unfair. Overall, the rebuttal partially addressed the concerns, but not strong enough. Therefore, the AC would like to recommend rejection by the end.

**Additional Comments On Reviewer Discussion:**

The authors have done a good job of rebuttal. During rebuttal and reviewer discussion, actually 2 of the 4 reviewers have increased their scores. Specifically,

1. Several reviewers have mentioned that the paper writing needs to be improved, and the paper is hard to follow at the first glance.

2. More experiments are needed to answer questions like the generalization of the proposed method, ablation of the reading and writing layers, more LLMs need to be tested, fairness of the comparison, etc. The authors have tried to address them in the rebuttal. But the AC also agree that more comprehensive experiments and analysis are still needed.

3. Several reviewers also questioned that it seems that for each target LLM, one has to train a model to decode its activations. So, a model trained on one target LLM cannot be used to interpret activations in another target LLM, which is less ideal.

---

### Decision · Program_Chairs · 2025-01-22

Reject